# Discovery of a Natural Product That Binds to the *Mycobacterium tuberculosis* Protein Rv1466 Using Native Mass Spectrometry

**DOI:** 10.3390/molecules25102384

**Published:** 2020-05-21

**Authors:** Ali R. Elnaas, Darren Grice, Jianying Han, Yunjiang Feng, Angela Di Capua, Tin Mak, Joseph A. Laureanti, Garry W. Buchko, Peter J. Myler, Gregory Cook, Ronald J. Quinn, Miaomiao Liu

**Affiliations:** 1Griffith Institute for Drug Discovery, Griffith University, Brisbane, Queensland 4111, Australia; ali.elnaas@griffithuni.edu.au (A.R.E.); jianying.han@griffithuni.edu.au (J.H.); y.feng@griffith.edu.au (Y.F.); a.dicapua@griffith.edu.au (A.D.C.); t.mak@griffith.edu.au (T.M.); r.quinn@griffith.edu.au (R.J.Q.); 2Institute for Glycomics, Griffith University, Gold Coast, Queensland 4222, Australia; d.grice@griffith.edu.au; 3Physical and Computational Sciences Directorate, Pacific Northwest National Laboratory, Richland, WA 99354, USA; joseph.laureanti@pnnl.gov; 4Earth and Biological Sciences Directorate, Pacific Northwest National Laboratory, Richland, WA 99354, USA; garry.buchko@pnnl.gov; 5School of Molecular Biosciences, Washington State University, Pullman, WA 99164, USA; 6Center for Global Infectious Disease Research, Seattle Children’s Research Institute, Seattle, WA 98109, USA; peter.myler@seattlechildrens.org; 7Department of Microbiology and Immunology, University of Otago, Dunedin 9016, New Zealand; greg.cook@otago.ac.nz

**Keywords:** altholactone, tuberculosis, Rv1466, drug target, native mass spectrometry

## Abstract

Elucidation of the mechanism of action of compounds with cellular bioactivity is important for progressing compounds into future drug development. In recent years, phenotype-based drug discovery has become the dominant approach to drug discovery over target-based drug discovery, which relies on the knowledge of a specific drug target of a disease. Still, when targeting an infectious disease via a high throughput phenotypic assay it is highly advantageous to identifying the compound’s cellular activity. A fraction derived from the plant *Polyalthia* sp. showed activity against *Mycobacterium tuberculosis* at 62.5 μge/μL. A known compound, altholactone, was identified from this fraction that showed activity towards *M. tuberculosis* at an minimum inhibitory concentration (MIC) of 64 μM. Retrospective analysis of a target-based screen against a TB proteome panel using native mass spectrometry established that the active fraction was bound to the mycobacterial protein Rv1466 with an estimated pseudo-*K*_d_ of 42.0 ± 6.1 µM. Our findings established Rv1466 as the potential molecular target of altholactone, which is responsible for the observed in vivo toxicity towards *M. tuberculosis*.

## 1. Introduction

Tuberculosis (TB) is the leading cause of death by an infectious agent worldwide, claiming an estimated 1.3 million lives in 2018, with an estimated 10 million people becoming ill with the disease [1]. The etiological agent responsible for TB is *Mycobacterium tuberculosis* (*Mtb*), an airborne bacterial pathogen. Humans are the only known reservoir for *Mtb*, where a complex mechanism has evolved that allows the pathogen to survive and replicate within macrophages and maintain a life-long latency within the host [2]. While modern public health care systems and effective drug treatment programs have made TB “invisible” for the most part in North America and Western Europe, the emergence of multidrug and extremely drug-resistant strains of *Mtb* could allow the disease to return to this part of the world with a vengeance [3]. In 2018, there were half a million new cases of rifampicin-resistant *Mtb*, of which 78% were multidrug-resistant TB [1]. Consequently, there is an urgency to develop new intervention therapies to keep ahead of the disease. 

A rich potential resource for the development of new intervention strategies against TB and other infectious diseases are natural products [4,5]. Since ancient times, drugs derived from natural sources have been employed to treat diseases and ailments. Recently there has been a return to natural products for lead compound identification [6,7,8]. Natural products are largely secondary metabolites with small molecular weights extracted from other living organisms. The wide diversity of life under different environments provides a chemical diversity of low molecular weight compounds that do not exist in comparison to standard combinatorial organic chemistry. Given that less than 10% of nature has been evaluated for medicinal properties, natural products represent a large under-explored reservoir for drug discovery [9]. As this reservoir is rapidly shrinking due to human activities, such as poor land management [10], war [11], and global warming [12], reducing the biodiversity on our planet [13], there is an urgency to harness these natural resources for medical purposes before they disappear.

In contrast to the large number of compounds isolated from nature, the number of these compounds whose target molecules have to date been identified is limited. Elucidation of the mechanism of action of bioactive natural products that show antimicrobial activity towards infectious diseases is an important but challenging field [14]. At the same time, this is the major bottleneck for drug development of synthetic compounds identified in phenotype-based screening. Over the years, several new target identification strategies have been developed and the number of successful examples is steadily growing [15]. Furthermore, in the case of natural product drug discovery, the additional challenge is low compound availability, limiting the use of some target identification strategies.

## 2. Results

In this work, we used both phenotype-based screening and target-based screening to sequentially identify a low molecular weight compound with phenotypic activity along with the potential molecular target of the molecule (Figure 1). Identifying a putative protein target allows the initiation of a target validation campaign to understand the underlying mechanism of action [16]. With regards to targets with uncharacterized/unannotated functions, ligand identification can be a vital first step towards decoding the biochemical function because the identified ligand and natural substrate might share functional groups or structural features [17]. In this study, phenotypic activity was identified through high throughput screening of a natural product fraction library against *M. tuberculosis.* Library fractions with favorable phenotypic activity were then screened against a panel of purified putative *M. tuberculosis* targets. This was achieved with native mass spectrometry screening employing electrospray ionization magnetic resonance mass spectrometry (MRMS) [18,19]. A crude estimation of the ligand’s dissociation constant, *K*_d_, with the target could be obtained by a mass spectrometry (MS) dose response curve.

A high-throughput screening (HTS) of fractions from plant species and marine organisms against *Mtb* identified a series of active fractions. Fraction 3 derived from the plant *Polyalthia* sp. showed anti-*Mtb* activity with an MIC value of 62.5 μge/μL. Further isolation and purification led to identification of a known compound, altholactone, with a molecular weight of 232 Da (Figure 2A). Altholactone was first discovered from a *Polyalthia* species in 1977 [21]. Previous studies showed that altholactone exhibited antimycobacterial activity against the *Mtb* strains H_37_Ra and *Mtb* H_37_Rv, with MIC values of 6.25 μg/mL (26.9 μM) [22] and 32 μg/mL (137.9 μM) [23], respectively. In the present study, we extended these explorations to evaluate the antimycobacterial effect of altholactone against *M. smegmatis* mc^2^155 and *Mtb* mc^2^6230. As shown in Figure 2B, altholactone showed activity against *M. smegmatis* with an MIC value of 500 μM (tested range 3–500 μM). In addition, it showed activity against the *Mtb* strain mc^2^6230 (MIC = 64 μM, tested range 4–512 μM) that contained two deletions in the 9455-bp region of deletion 1 (RD1) of the protein in the pantothenate biosynthesis pathway, to render the organism non-pathogenic.

We have recently published a new natural product drug discovery approach ‘PhenoTarget Drug Discovery’, which combines phenotypic screening followed by target screening on the phenotypic active natural product fractions [24]. The objective was to identify compounds with cellular activities and also identify the molecular target. This approach could be equally applied to pure compound libraries as well as natural product fraction/extract libraries, due to the development of a highly sensitive and easy-to-perform target screening technique using cloned and purified proteins. The target identification component directly observes non-covalent and covalent protein–ligand complexes. A particular advantage of this method is that it requires no tag, e.g., a biotin or fluorescent tag to be added to the compounds or proteins. The rapid, label-free native MS approach depends on non-denaturing electrospray-ionization (ESI) to recognize multi-charged proteins in their near-native states. Due to the gentle conditions employed to spray the target in the native state it was possible to directly observe non-covalent and covalent protein–ligand complexes [25]. The difference between the mass-to-charge ratio (Δ*m*/*z*) for the protein–ligand complex and the unbound protein ions multiplied by the charge state (*z*) directly provided the molecular weight of the bound ligand (hit) (MW_ligand_ = Δ*m*/*z* × *z*). 

Retrospective analysis of native MS screening on 40 pool fractions combined with 362 hit fractions with anti-*Mtb* H_37_Rv cellular activity against a panel of 37 purified putative mycobacterial drug targets was conducted. A pool fraction containing a “hit” fraction derived from *Polyalthia* sp. was identified as forming a protein–ligand complex with the *Mtb* protein Rv1466. As shown in Figure 3A, three charge states, 5+, 6+, and 7+ were observed for Rv1466, under the native MS conditions. As shown in Figure 3B, upon addition of the active pool fraction, the same three charged states were observed but all shifted to higher *m*/*z*. Using the dominant peak as an example, the molecular weight of the bound ligand was calculated to be 232.06 Da ((2462.35069–2423.67327) × 6 **=** 232.06 Da), which was consistent with the covalent binding of the isolated compound altholactone (Figure 3B). Identification of altholactone was confirmed by the observation of the same mass spectrum using isolated pure altholactone and Rv1466 (Figure 3C).

Electrospray ionization-mass spectrometry (ESI-MS) has proven to be a useful tool for determining dissociation constants (*K*_d_) for a variety of biological noncovalent complexes [26,27]. For covalent binders, a pseudo-*K*_d_ can be determined. It is measured in protein–ligand systems by calculating the electrospray-ion abundances of the free protein and the complexes [28]. Using the automated nanoESI-MS, the pseudo-*K*_d_ of altholactone and Rv1466 was measured by the titration approach using a constant Rv1466 concentration and titrating in the altholactone. Increasing the amount of ligand in the mixture results in the increased formation of the complex. The changes in the ratio of free Rv1466 to bound Rv1466 was used to calculate the pseudo-*K*_d_.

Figure 4 shows fourteen native mass spectra of samples containing 4.5 μM Rv1466 and increasing concentrations of altholactone (0.001–1000 μM). A ligand concentration was reached (~1000 μM) where the intensity of the protein–ligand complex reached a plateau. The ratios of the intensity of the protein–ligand peak and sum of protein peak plus protein–ligand peak were plotted against the concentration of altholactone (Figure 4). Using these ratios and Equations (1) and (2), a pseudo-*K*_d_ of 42.0 ± 6.1 µM was calculated for altholactone binding to Rv1466. 

The structure of Rv1466 was solved by SSGCID using NMR-methods and the ensemble of structures can be accessed from the Protein Data Bank (5IRD). To assess the binding of altholactone to Rv1466, altholacone was docked onto the structure closest to the average structure in the ensemble of NMR structures, using the program PDB2PQR [29] via UnityMol [30] to prepare the structures at pH 6.9. AutoDock Vina [31] was used within Chimera [31] to predict non-covalent binding of the altholactone within the Rv1466 pocket. The structure shown in Figure 5 is the lowest energy structure with a score of −6.7 kcal/mol, a value that falls within a linear correlation between the predicted and experimental values, giving high confidence to the docking experiments [31]. As illustrated in Figure 5A, the docking program suggests that the altholactone bound to a pocket in the protein formed by a three-strand β-sheet and two parallel α-helices. As shown in the surface rendition in Figure 5B, the altholactone bound deep inside this pocket. The mass spectrometry data suggests that a Ser, Thr, or Lys residue might form a covalent bond with the altholactone, and, as illustrated in Figure 5, the side chain of S9 sits close to the ligand where a reaction could occur.

## 3. Discussion

In this work, a known compound altholactone was isolated from the plant *Polyalthia* sp. Altholactone has demonstrated anti-inflammatory and anticancer activities against eukaryotic tumor cells, and antifungal/antimicrobrial activity against both gram positive and negative bacteria [32,33,34,35], but the molecular mechanisms responsible were still not fully defined. Studies have also shown altholactone cytotoxicity against human HeLa cells with an IC_50_ of 148.7 μM [23]. Mechanism studies reported that altholactone inhibited the growth of human bladder cancer T24 cells (IC_50_ = 43.5 μM) by inducing apoptosis [35] and induced DU145 cell death (IC_50_ = 38.5 μM), through inhibition of NF-κB and STAT3 activity [34].

In the current study, altholactone was shown to inhibit pathogenic *Mtb* and the fast growing non-pathogenic *M. smegmatis*, with MIC values of 64 and 500 μM, respectively. The lack of congruence between results obtained from the *Mtb* and *M. smegmatis* assay was observed and discussed extensively. While using *M. smegmatis*-based screens for anti-mycobacterial drug discovery remains well-recognized, an important limitation of *M. smegmatis* as a surrogate organism for *Mtb* drug discovery was sensitivity [36]. A significant number of molecules identified in *Mtb* assays did not show activity against *M. smegmatis* and, thus, would have been missed [37].

Retrospective analysis of a previous target screening study of a natural product fraction library against a panel of 37 *Mycobacterium* proteins by native MS identified a binding complex between Rv1466 and a ligand with a molecular weight of 232.06 Da. In the screening of 362 active fractions, only six fractions containing something that bound to Rv1466, ruling out the possibility of promiscuous small molecule binding to Rv1466. Further native MS experiments confirmed the ligand in one fraction as altholactone. Rv1466, along with proteins Rv1460 to Rv1465, comprise the primary Fe–S cluster assembly and repair SUF (mobilization of sulfur) machinery in *Mtb* [38]. Investigation of small molecule inhibitors of SUFs in various pathogenic models have been extensively conducted, such as D-cycloserine binding to *Plasmodium falciparum* SufS (29.2 ± 2.9 µM) [39], and VU0038882 binding to the *Staphylococcus aureus* iron-sulfur (Fe–S) cluster (2.1 µM) [40]. A high-density mutagenesis study confirmed that six genes, Rv1461 to Rv1466, were required for in vitro mycobacterial growth [41]. It was reported that the SUF system plays an essential function for *Mtb* survival due to its role in providing bacterial resistance to iron limitation and oxidative stress [42]. Interruptions of individual proteins (Rv1461, Rv1462 or Rv1463) led to impeding mycobacterial growth [38], suggesting the high possibility that inhibiting a component of the multiprotein SUF complex would affect the whole SUF system, and eventually, lead to mycobacterial death.

Altholactone is likely to undergo an irreversible hetero-Michael addition reaction with an amino acid, such as serine, threonine, or lysine residue of Rv1466. Covalent inhibitors possess advantages such as enhanced biochemical efficiency, improved pharmacokinetics properties, and potential to overcome drug resistance [43,44]. While a protein–ligand complex between altholactone and Rv1466 was identified by our mass spectral assay that suggested a reaction with the side chain of a serine, threonine, or lysine group, the observation that none of the other 36 proteins in our screen reacted with the active fraction or pure altholactone suggests the compound does not react promiscuously with all proteins. Our molecular modelling indicates that a specific ligand binding pocket on the surface of Rv1466 exists where such a covalent reaction could occur. Further studies, including the use of Rv1466 knock-out strains of *Mtb*, are necessary to confirm if inhibition of Rv1466 by altholactone is the mechanism of action.

Our findings support that altholactone could represent a novel chemotherapeutic natural agent or lead scaffold against tuberculosis. Since the concentration of altholactone required to inhibit mycobacterial species is rather high, more potent analogs of altholactone need to be developed for practical use in TB treatment.

## 4. Materials and Methods 

### 4.1. General Experimental Procedures 

NMR spectra for altholactone were recorded in DMSO-*d_6_* (δ_H_ 2.50 and δ_C_ 39.5) at 25 °C on a Bruker AVANCE III HDX 800 MHz NMR spectrometer (Fallanden, Zurich, Switzerland) equipped with a triple resonance cryoprobe. High-resolution electrospray ionization mass spectra (HRESIMS) were recorded on a Bruker maXis II ETD ESI- qTOF (Bruker, Bremen, Germany). The HPLC system for LLE fractions for phenotypic screening included a Waters 600 pump (Milford, MA, USA) fitted with a 996-photodiode array detector and Gilson FC204 fraction collector (Middleton, WI, USA). The HPLC system for purification of re-extracted material was a semi-preparative Thermo Ultimate 3000 system with a PDA detector (Waltham, MA, USA). A Phenomenex C18 Monolithic column (5 μm, 4.6 × 100 mm) was used for LLE fractionation; a Thermo Electron Betasil C18 column (5 μm, 21.2 × 150 mm) was used for semi-preparative HPLC.

### 4.2. Natural Product Fraction Library 

The lead-like enhanced natural product fraction library (202,983 fractions) was constructed as previously described [20]. 

### 4.3. Phenotypic Screening against M. tuberculosis H_37_Rv 

Growth inhibition of *M. tuberculosis* H_37_Rv using the natural product fraction library was conducted as previously reported [24]. Fractions were prepared from dried, ground biota, as previously described [20]. The terminology 250 μge/μL meant the fraction originated from 250 μg of dried, ground biota. Fractions were prepared to the concentration of 250 μge/μL and growth inhibition of the *M. tuberculosis* strain H_37_Rv screening was monitored using 1 μL fraction, followed by serial dilution in 384-well plates. To this, 40 μL of *M. tuberculosis* ATCC 27294 H_37_Rv (3–5 × 10^5^ CFU/mL in Middlebrook 7H9 broth with 0.05% Tween 80, 10% *v*/*v* ADC and Casamino acids) was added with a Multidrop dispenser. The plates were then incubated at 37 ℃ for 7 days. A 10 μL solution of Resazurin (20 mg/100 mL diluted 1:1 with 10% Tween 80) was added and incubated further for an additional 24 h at 37 °C for color development. Absorbance was monitored at two wavelengths (575 and 610 nm) using Spectramax and the ratios were determined to calculate the % inhibition. Growth controls in the absence of compound as well as media controls served as inhibition ~0% and −100%, respectively.

MIC—the least concentration which gave ≥80% inhibition was considered as MIC (start conc. = 1 μL of fraction (250 μge/μL).

### 4.4. Re-Extraction and Purification

Freeze-dried *Polyalthia* sp. (10.0 g) was sequentially extracted with *n*-hexane (250 mL), CH_2_Cl_2_ (250 mL), and MeOH (2 × 250 mL). The CH_2_Cl_2_ and MeOH extracts were combined and dried under reduced pressure, to yield a dark brown solid (1.50 g). The plant crude extract was pre-adsorbed to a roll of cotton and then packed into a stainless-steel cartridge (10 × 30 mm) that was subsequently attached to a C18 Betasil HPLC column (250 × 21.2 mm). Isocratic HPLC conditions of 90% H_2_O (0.1% TFA)/10% MeOH (0.1% TFA) were employed for the first 10 min, and then a linear gradient to MeOH (0.1% TFA) was run over 60 min, followed by isocratic conditions of MeOH (0.1% TFA), for a further 10 min, all at a flow rate of 9 mL/min. Sixty fractions (60 × 1 min) were collected from the start of the HPLC run. The ^1^H NMR 800 MHz fingerprints of fraction 30 compared to the ^1^H NMR of the active fraction confirmed the presence of altholactone (10 mg, dry weight).

### 4.5. Altholactone 

Brown solid; HRMS *m*/*z* 255.0623 [M + Na]^+^, *m*/*z* 487.1353 [2M + Na]^+^, calcd. MW for 232.0623. ^1^H NMR (DMSO-*d_6_*, 800 MHz) *δ*_H_ 7.34 (2H, m, H-3′,5′), 7.28 (3H, m, H-2′, 4′, 6′), 7.13 (1H, dd, J = 9.8, 4.9 Hz, H-7), 6.20 (1H, d, J = 9.8 Hz, H-6), 4.89 (1H, dd, J = 5.0, 2.2 Hz, H-4), 4.67 (1H, d, J = 4.9 Hz, H-2), 4.64 (1H, t, J = 5.2 Hz, H-8), 4.11 (1H, dd, J = 5.2, 2.2 Hz, H-3); ^13^C NMR (DMSO-*d_6_*, 200 MHz) *δ*_C_ 160.97 (CO), 141.32 (CH, C7), 139.53 (C, C1), 128.36 (CH, C3′, 5′), 127.72 (CH, C4′), 125.88 (CH, C2′, 6′), 123.11 (CH, C6), 85.88 (CH, C2), 85.78 (CH, C4), 83.02 (CH, C3), 67.97 (CH, C8).

### 4.6. Biological Assays 

Altholactone was evaluated for its anti-mycobacterial activities against the *M. smegmatis* strain mc^2^155 (ATCC 70084) and *M. tuberculosis* mc^2^6230. The *M. smegmatis* strain mc^2^155 was grown in Middle brook 7H9 broth (Difco, Sparks, MD, USA) supplemented with 10% (*v*/*v*) OADC enrichment (Becton Dickinson) 0.05% (*v*/*v*) Tween-80 and 0.2% (*v*/*v*) glycerol. Cultures were grown at 37 °C, while shaking (200 rpm). In clear-bottomed 96-well plates (Nunc), two-fold serial dilutions (8 times) of each compound were added to volumes of 40 μL of 7H9 medium. The last column of each plate did not contain any compound and served as a negative control. Previously prepared *M. smegmatis* strain mc^2^155 inocula were diluted with their culture medium to achieve 0.0025 OD_600_. Isoniazid served as a positive control and DMSO as a negative control. The plate was incubated at 37 °C for 24 h. After incubation, 30 μL of 0.02% resazurin was added to the wells, and the plates were incubated at 37 °C for 4 h. The minimal inhibitory concentration was determined by the compound concentration well that remained blue and did not change to pink. All experiments were performed in triplicates.

The *M. tuberculosis* strain mc^2^6230 used in this study was obtained from the Howard Hughes Medical Institute, Department of Microbiology and Immunology, Albert Einstein College of Medicine. Mycobacterial strains were grown in Middlebrook 7H9 medium (Difco, Sparks, MD, USA) supplemented with 10% (*v*/*v*) OADC enrichment (Difco), 0.2% (*v*/*v*) glycerol, 0.05% (*v*/*v*) tyloxapol, and pantothenate (50 mg/L). Cultures were grown at 37 °C, while shaking (160 rpm). In clear-bottomed 96-well plates (Nunc), two-fold serial dilutions (8 times) of each compound were added to volumes of 100 μL of 7H9 medium. The last column of each plate did not contain any compound and served as a negative control. Previously prepared *Mtb* mc^2^6230 inocula were diluted with their respected medium to achieve 0.05 OD_600_. The inocula were then added to each well and the plates were incubated at 37 °C for five days. After incubation, 30 μL of 0.02% resazurin was added to the wells, and the plates were incubated at 37 °C for 24 h. The minimal inhibitory concentration was determined by the compound concentration well that remained blue and did not change to pink. All experiments were performed in triplicates.

### 4.7. Target Screening 

Rv1466 and the other 36 unique proteins (Figure 6) in the target panel were supplied by the Seattle Structural Genomics Center for Infectious Diseases (SSGCID, Seattle, WA, USA; www.ssgcid.org). Target screening of the pool fraction library against the protein panel was conducted as previously reported [24]. When the protein-ligand complex was found, the molecular weight of the binding ligand was estimated from the spectrum, using the following equation: MW ligand = ∆*m*/*z* × *z*.

### 4.8. Pseudo-K_d_ Determination 

Altholactone solutions were prepared in DMSO through serial dilution (0.01 µM, 0.03 µM, 0.1 µM, 0.3 µM, 1 µM, 3 µM, 10 µM, 30 µM, 100 µM, 300 µM, 1 mM, 3 mM, 6 mM, 10 mM). Each concentration (1 µL) was added to each well of a V-plate microtiter plate (BioCentrix, Carlsbad, CA, USA). The DMSO in each well was dried off using a Freeze dryer (Christ, Osterode am Harz, Germany), followed by the addition of 1 µL of MeOH to each well. Rv1466 was buffer-exchanged into ammonium acetate (500 mM, pH 6.9) using a Nalgene NAP-5 exclusion column, prior to ESI–MS analysis. Rv1466 (9 µL) was added to each well containing the altholactone. Samples were incubated for 30 to 60 min at room temperature. All sample solutions were injected by fully automated chip-based nanoelectrospray. The experiment was performed in triplicate. 

The relative abundances of the protein–ligand complex to total protein in the mass spectra correlated to the relative equilibrium concentrations of the ligand to the total protein in solution. The pseudo-*K*_d_ of altholactone with Rv1466 was determined using the following equations:(1)$$\frac{\sum I{\left(P-L\right)}^{n+}/n^{\;}}{\;\sum I{\left(P\right)}^{n+}/n^{\;}+\sum I{\left(P-L\right)}^{n+}/n^{\;}}=\frac{\left[P-L\right]}{{\left[P\right]}_{t}}$$
(2)$$\frac{\sum I{\left(P-L\right)}^{n+}/n^{\;}}{\;\sum I{\left(P\right)}^{n+}/n^{\;}+\sum I{\left(P-L\right)}^{n+}/n^{\;}}=\frac{{\left[P\right]}_{t}+{\left[L\right]}_{t}+\mathrm{Kd}-\sqrt{{\left({\left[P\right]}_{t}+{\left[L\right]}_{t}+K_{d}\right)}^{2}-4{\left[P\right]}_{t}{\left[L\right]}_{t})}}{2{\left[P\right]}_{t}}$$

Experimental relative ratios of the protein-ligand complex and total protein ion abundances were plotted against the total concentration of ligand.

## Figures and Tables

**Figure 1 molecules-25-02384-f001:**
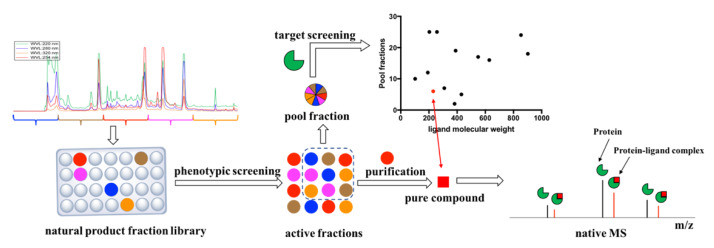
Overview of natural product drug discovery using both phenotypic screening and target screening approaches. A high throughput phenotypic screening of 202,983 Nature Bank (NB) lead-like enhanced (LLE) fractions [20] against *M. tuberculosis* H_37_Rv was initially performed. Active fractions were screened against a panel of 37 putative anti-tuberculosis (TB) targets from *Mycobacteria* species. To lower sample consumption, especially protein, nine active fractions were pooled (40 Pool Fractions) and incubated with each of the target proteins. The pooled fraction–protein mixtures were examined by native mass spectrometry to identify protein–ligand complexes. The mass shift between the protein (black) and the protein–ligand complex (red) peaks provided the molecular weight of the bound ligand.

**Figure 2 molecules-25-02384-f002:**
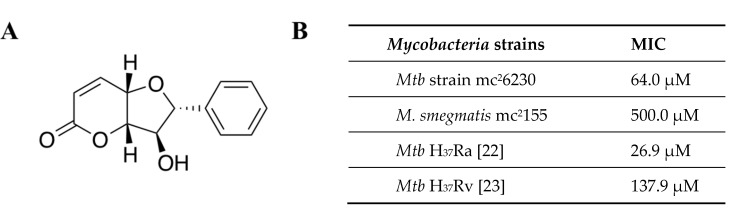
(**A**) Structure of altholactone; and (**B**) biological activity of altholactone against *Mycobacteria* strains and species.

**Figure 3 molecules-25-02384-f003:**
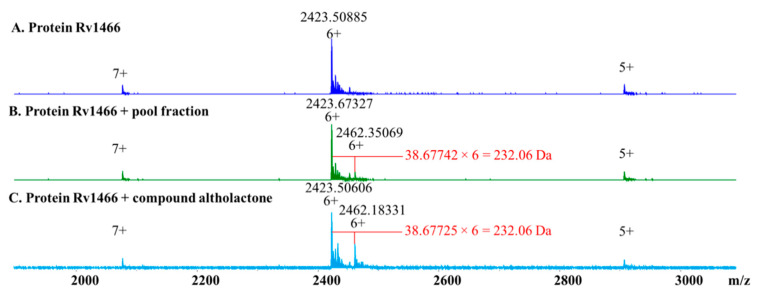
Native MS spectra of (**A**) Rv1466; (**B**) Rv1466 mixed with a pool fraction; and (**C**) Rv1466 mixed with pure isolated altholactone. The same binding ligand with molecular weight 232.06 Da was identified from both the pool fraction and pure compound altholactone.

**Figure 4 molecules-25-02384-f004:**
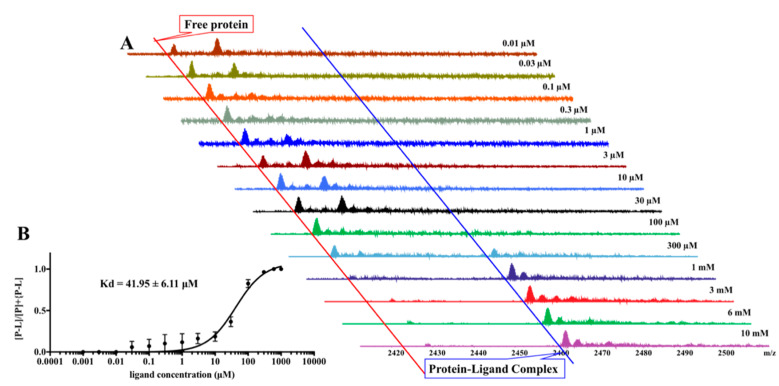
Determination of the pseudo-*K*_d_ between altholactone and Rv1466. (**A**) Overlay of the 14 mass spectra of Rv1466 at a concentration of 4.5 μM mixed with increasing concentration of pure altholactone (0.001–1000 μM). (**B**) Plot of [P-L]/[P] + [P-L] versus ligand concentrations for the titration of Rv1466 with altholactone. The pseudo-*K*_d_ was calculated as 42.0 ± 6.1 μM.

**Figure 5 molecules-25-02384-f005:**
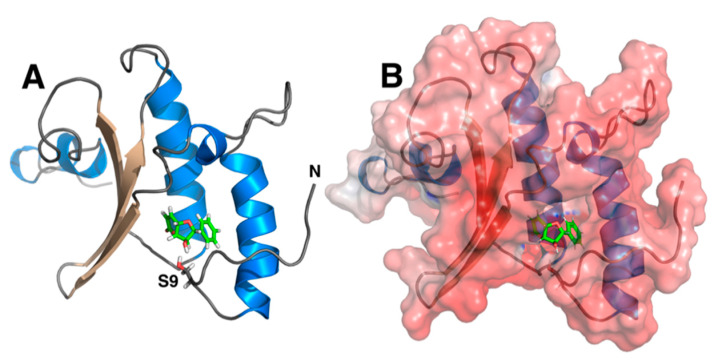
(**A**) Cartoon representation of the structure closest to the average structure of an ensemble of solution-state NMR structures calculated for Rv1466 (5IRD). Docking experiments show that the altholactone bound into a pocket formed by a three-strand β-sheet (pale) and two parallel α-helices (blue). The atoms of the stick representation of altholactone are colored green (carbon), red (oxygen), and white (hydrogen). The side chain of a serine residue, S9, near the altholactone is highlighted. (**B**) Surface renditions of the single structure shown in Figure 5A with a stick representation of altholactone.

**Figure 6 molecules-25-02384-f006:**
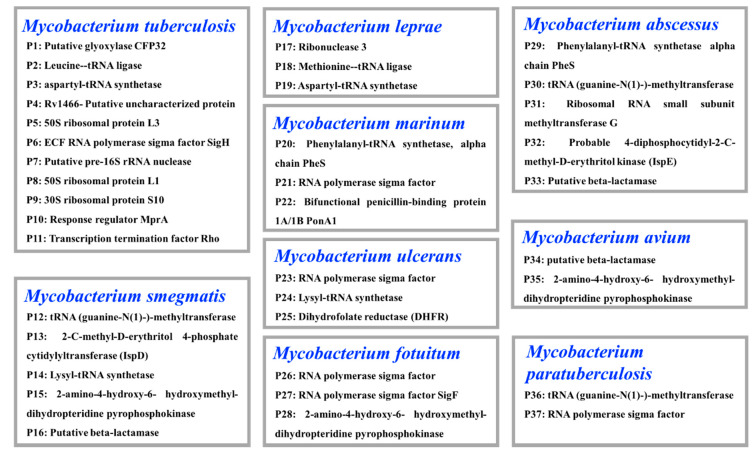
List of the collection of 37 TB proteins from nine *Mycobacteria* species.

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
