# Peer review of "Discovery of a Natural Product That Binds to the Mycobacterium tuberculosis Protein Rv1466 Using Native Mass Spectrometry"

_molecules, 2020, doi:10.3390/molecules25102384_

Round 1

Reviewer 1 Report

This is a very nice manuscript outlining the discovery of a new potential molecule for the treatment of TB. Although the MIC is rather on the higher end, being in the upper µM range, it might a good potential being a small molecule. Nice figures outlining the procedures of drug discovery as well as determining Kd. Overall, the manuscript makes a good impression with minor text editing required.  

Reviewer 2 Report

The manuscript “Discovery of a specific binding ligand of Mycobacterium tuberculosis protein Rv1466 by native mass spectrometry” outlines the identification a small molecule inhibitor (altholactone) of Mycobacterium tuberculosis, through a natural product screen and identification of the protein target (Rv1466) through native ESI-MRMS. The authors present MIC data not only for M. tuberculosis H37Rv, but also other M. tuberculosis strains as well as Mycobacterium smegmatis, showing a greater activity of altholactone against pathogenic mycobacteria. This work is interesting, and the manuscript is well written and has a solid structure.

However, there are several minor points to be revised.

The figure legend for figure 1 requires more detail. As it stands, the specific steps involved in the approach are unclear.

Line 46: Mtb should be italicised for consistency.

Line 71: “sequentially identify a low molecular weight with phenotypic activity”, should this be low molecular weight compound? Please clarify.

Line 92/93: “effect of altholactone against M. smegmatis and Mtb.” Please clarify which strains of Mtb. Also “and” should not be italicised.

Line 107: “A particular advantage of this method is that it requires no any tag…” Please clarify.

Line 153: “library against a panel of 37 Mycobacterium proteins”, Mycobacterium should be italicised.

Line 160/163: Rv1460 is referred to in the text, should this be Rv1466? Please clarify.

Line 198/199: Should match the style of Line 208/209 (or vice versa) for consistency.

Line 208/209: Resazurin is used in the assay with M. tuberculosis strain mc26230 for MIC determination. Why is resazurin not used for the assay with M. smegmatis mc2155?

In view of the above comments, I recommend that the article be accepted upon addressing each point.

Reviewer 3 Report

The authors have carried out a phenotypic and target based screening approach to identify molecules from natural resources that can be used to develop drugs that can target MDR-TB strains.

The work is well developed and presented.

A couple of minor points:

  1. Results: Line 93: The authors should specify the concentration range of altholactone used to determine MIC in Figure 2B. 
  2. Line 107: "no any tags" to "no tags"
  3. Methods: Biological assays: 2 fold serially diluted how many times? this will give a reader a sense of the dilution series used.
  4. The references should be inserted before the full-stop or comma in the relevant areas.

Reviewer 4 Report

Summary:  This manuscript reports the identification of a known plant-derived natural product, altholactone, in a phenotypic high-throughput screen against Mycobacterium tuberculosis followed by identification of a putative target protein using a native mass spectrometry based approach.  This represents the second report of the use of this so-called “PhenoTarget Drug Discovery strategy” to address this important bottleneck in antimicrobial drug discovery, and NP drug discovery in particular.  Validation of this approach exploiting this powerful biochemical tool to rapidly identify putative targets of active extracts or pure NP compounds is a significant contribution to the field.  It was well-written with the experimental approach and data clearly presented.  The main areas of concern include the low potency of the compound, the lack of consideration of cytotoxicity/selectivity, the lack of evidence that Rv1466 is actually inhibited by altholactone or that the observed binding is specific. These issues and minor revisions are detailed below.

Major issues:

  1. There is insufficient data to support the claim that Rv1466 is THE molecular target of altholactone (line 33).
    1. Only ~40 putative targets were evaluated
    2. The specificity of binding is not known. How many other extracts/fractions showed binding to Rv1466?  There is at least one prior paper by this group of another compound that binds to this target, raising the possibility of promiscuous small molecule binding. 
    3. The data only show weak binding to Rv1466, but inhibition of the target is not shown.  Likewise, no corroborating data derived from other experimental approaches (i.e. selection of spontaneous mutants, overexpression of putative targets) is provided. 
    4. Impact of assaying for binding to individual protein that is normally a component of multiprotein SUF complex is unknown and should at least be discussed.
    5. Line 160 – The proposal that altholactone would form covalent bonds with Ser, Thr, or Lys residues would not seemingly allow for specific target interactions but would rather lead to interaction with any protein. Please clarify. 
    6. A crystal structure for the putative target is available – Virtual docking experiments may (or may not) substantiate this central claim.
  2. The potency of this compound in whole cell assays does not meet standard criteria for hit-to-lead progression, thus calling into question the value of additional efforts to identify the target. The lack of information regarding the cytoxicity/selectivity of altholactone should also be addressed to aid go/no-go decision making regarding this hit compound.  Is it known whether it exerts bacteriostatic or –cidal activity against Mtb?
  3. The reported pseudo Kd seems very high for a drug-target interaction. For reference the Kd of the front-line drug isoniazid (INH) in the form of the active INH-NAD adduct for binding to InhA is 16nM (step 1) and 0.75nM (step 2) (Rawat et al, PNAS Nov 25, 2003).  Please comment on the observed Kd for Rv1466-altholactone as it relates to known drug-target interactions in clinical use.

Minor Issues:

  1. Line 29 (and other places) – Should mge/mL be mg/mL? If not, why is an “e” included only in some usages?  Likewise, please confirm that this should be mg/mL rather than mg/mL. 
  2. Line 40 – TB does not really afflict 1/3 of the global population. The vast majority of this group has LTBI and is unaware they are infected.  Suggest modifying the wording. 
  3. Line 71 – A word seems to be missing. Should read “identify a low molecular weight compound’
  4. Line 74 – It is stated that ligand identification is vital first step to elucidating biochemical functions of uncharacterized targets.  While identifying native biologically relevant ligands may serve this role, it is unclear how identifying drug/NP ligands would shed light on target protein functions.  Please clarify.
  5. line 93 – The relevance of the activity against non-pathogenic smeg is unclear. Published studies have previously demonstrated that lack of congruence in susceptibility of Msmeg and Mtb.
  6. Line 96-97 – “smegamatis” is misspelled in Fig 2B.
  7. Lines 115-118 – If each pool fraction used for MS screening contains constituents from ~9 extract samples, how is it determined that the ligand originates from a specific Hit fraction from Polyalthia? This aspect of the workflow was unclear from the text. 
  8. Line 116 – The identity of the other 36 protein drug targets used in the screen should be described in the methods. Also, how was the integrity/functionality of the proteins verified?  Factors such as stability, solubility, aggregation could impact the results of the MS screen.
  9. In the Discussion, inclusion of citations of examples of small molecule inhibitors of the SUF pathway would help to establish the context of this study. Based on cursory literature search, there are a number of previous SUF inhibitors. 
  10. Line 156 – I believe this should read “Rv1460-1465 comprises the primary…” or “the Fe-S cluster assembly consists of Rv1460-65.
  11. Line 160 and 163 mention Rv1460. This should be Rv1466, if I am not mistaken.
  12. Line 177 – Regarding the HTS assay, please indicate the signal/background ratio and Z’ of the assay.
  13. Line 235 – The reference to polycarpine (ligand of Rv1466 reported in previous paper) appears to be in error.

Round 2

Reviewer 4 Report

The authors were very responsive to previous critiques, resulting in an improved manuscript.  They were very thorough in either making suggested changes or explaining/clarify any points in question.